# Deep Learning-based Discrimination of Pause Episodes in Insertable Cardiac Monitors

## Abstract

Remote monitoring of patients with insertable cardiac monitors (ICMs) has revolutionized follow-up procedures and enhanced the timely diagnosis of cardiac arrhythmias. Despite these advancements, challenges persist in managing and adjudicating the data generated, placing strain on clinic resources. In response, various studies have explored the application of Convolutional Neural Networks (CNNs) to classify raw electrocardiograms (ECGs).

The objective of this study was to create and assess a CNN tailored for the reduction of inappropriate pause detections in ICMs. A customized end-to-end CNN model comprising 5 convolutional layers for rhythm classification of ICM-detected pause episodes was developed. The training data consisted of ICM-detected pause episodes from 1,173 patients. After training the model, we evaluated its performance using a test dataset of ICM-detected pause episodes from 750 independent patients. All pause episodes utilized in training and testing were adjudicated manually as either true or false detection. The training dataset consisted of 4,308 pause episodes (2,722 true episodes from 960 patients and 1,586 false episodes from 251 patients). The validation dataset includes 1,095 detected Pause episodes from 256 patients (677 true pause from 203 patients and 418 false pause episodes from 58 patients) and had an area under the curve (AUC) of 0.994 for the proposed CNN. The optimal threshold was chosen to obtain 99.26% sensitivity and 96.89%. The test dataset consisted of 1,986 episodes (744 true episodes from 382 patients and 1,242 false episodes from 485 patients. The model demonstrated an AUC of 0.9942, 99.06% sensitivity, and 95.17% specificity in the test dataset.

The customized CNN model, 737 out of 744 episodes were correctly identified as pauses, resulting in a positive predictive value of 92.47%. Consequently, there was a reduction of EGM burden by 59.87%.

## 1 Introduction

Subcutaneous implanted insertable cardiac monitors (ICMs) have gained widespread use in clinical practices for rhythm diagnosis and management of cardiac arrhythmias in patients with unexplained syncope, palpitations, cryptogenic stroke, and following ablation procedures Brignole et al. (2018); Al-Khatib et al. (2018). These devices offer patient-triggered electrocardiogram (ECG) storage and automatic detection of various cardiac arrhythmias, including pauses, bradycardia, tachycardia, and atrial fibrillation, with high sensitivity. The miniaturization of ICMs, simplified implant procedures, enhanced automation, and wireless data transmission to patient care networks have significantly improved their acceptance among physicians and patients.

Long term follow-up and remote monitoring produces a substantial volume of ECG's estimated at $> 100$ million per year in the U.S. alone Tison et al. (2019). False pause detection, primarily due to QRS wave undersensing, presents a common challenge in the existing ICM systems. These falsely triggered episodes cause additional review burden and resource utilization since device clinics need to go through all device detected episodes to identify clinically important arrhythmias. This manual review and interpretation of ECG data is repetitive, tedious, and time-consuming, which leads, in some cases, to misclassifications and undiagnosed patients. As the use of ICMs continues to increase and monitoring durations extend, minimizing false pause detections has become increasingly important.

In addressing this challenge, we have harnessed the power of deep learning models, specifically convolutional neural networks (CNNs), which have demonstrated promising results in the realm of ECG classification Ali et al. (2022); Liu et al. (2021); Huang et al. (2023); Ebrahimi et al. (2020). These CNN models are designed to directly process the raw ECG signal, enabling the automatic extraction of pertinent features as the deep learning network refines its capabilities over multiple iterations and epochs. Moreover, our innovative approach involves the deployment of a cloud-based solution, affording all users of ICM devices immediate access to algorithmic enhancements as soon as they are released. This seamless integration ensures that our users always have access to the most current and precise algorithms, elevating the overall performance and dependability of our devices.

## 2 METHOD

### 2.1 PAUSE DETECTION IN ICM

Pause detection in ICMs is primarily based on sensing R-waves and detecting RR intervals longer than clinician programming of actionable pause duration. However, deciding based on sensed RR intervals alone is prone to inappropriate pause detection due to scenarios of undersensing changing R-waves peak amplitudes (e.g., body postural changes, heart axis deviations, varying tissue-electrode contact including loss of contact, ECG noise and motion artifact affecting signal baseline and signal saturation). Additional discriminators are used to further improve the accuracy of pause episode detection including secondary evaluation using fine-tuned sensing threshold to detect under-sense or smaller amplitude R-wave peaks, ECG baseline wandering detection noise detection. Once an episode is detected and confirmed by additional discriminators, a number of seconds of prior and preceding the pause event are stored as well as supplementary diagnostics and contextual information such as episode duration/time of day, posture/activity/AF status, etc. Device-based algorithms are limited in terms of complexity given constraints on size and battery capacity.

Recent advancements in device capabilities enable temporaneous transmission of stored episodes via Bluetooth technology to smartphone apps which in turn relay data to remote monitoring cloud servers and services capable of more complex computations and processing to organize streams of data from ICMs, alert clinics about episodes, and generate reports for clinic review.

### 2.2 DEEP LEARNING−BASED EPISODE CLASSIFICATION

The proposed CNN model consists of 5 convolutional layers, batch normalization, max pooling, dropout, global average pooling along with one fully connected and Softmax layer Figure 1. Model development and training were done using Python 3, TensorFlow v. 2.0, and Keras on a Windows 10 workstation with a Nvidia RTX 2080 GPU, Intel Xeon W-2125 CPU, and 32GB of RAM.

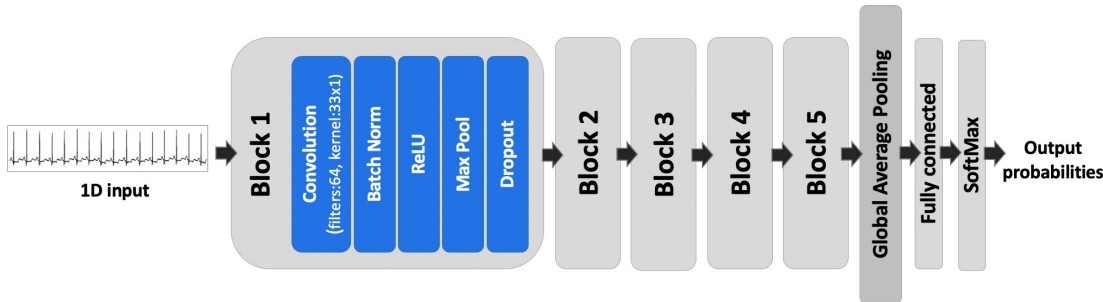

Figure 1: The basic architecture of the proposed CNN with 5 blocks of convolution layers, global average pooling, fully connected, and SoftMax.

## 3 RESULT

### 3.1 DATASET

A total of 4,308 detected Pause episodes from 1,173 patients (2,722 true pause episodes from 960 patients and, 1,586 false pause episodes from 251 patients) were used to train the CNN from initial random weights. The validation dataset had 1,095 detected Pause episodes from 256 patients (677 true pause from 203 patients and 418 false pause episodes from 58 patients). The validation set receiver-operating characteristic (ROC) curve was used to choose a probability threshold for classification into Pause vs. non-Pause episodes. Furthermore, to evaluate the performance of our proposed CNN, we conducted comparisons between the dense neural network (NN), Conv Net-Dense NN, Conv Net-GAP. The hyperparameters for each model such as activation, learning rate, kernel size, optimizer, regularization etc. are tuned accordingly.

The three proposed architectures were trained on the training dataset. Then, to evaluate the architecture, we checked the validation accuracy scores for different thresholds. We then selected the network architecture with the highest average accuracy score for all the thresholds. All networks were trained for 1000 epochs using an early stopping condition—i.e., when the accuracy on the validation set did not improve for 25 epochs, training stopped. All models were trained using ten-fold cross-validation. This cross-validation procedure requires a given model to be trained ten distinct times (re-initializing the network parameters each time) and ensures that, on the one hand, different subsets of the data are used for training and testing, while on the other hand, each data point serves as part of the training set (nine times) and in the test set (once). To be clear, when we performed cross-validation, we used data partitions that were not used during the hyperparameter search.

Moreover, increasing the size of the training dataset enhances the generalizability of any trained network. Data augmentation was employed with a magnification factor of 2, taking into account possible variations in the ICM device's positioning and movement, which could result in the presence of negative R-peaks in the dataset. To ensure transparency and reproducibility, the study reports results both with and without data augmentation for all baseline models, aiming to improve the quality and reproducibility of work in the field of data augmentation on deep learning-based ECG analysis. Figure 2 compares the performance metrics of sensitivity, specificity, accuracy, and F1-score for different models: baseline models proposed CNN (Conv Net-GAP), and variations with and without data augmentation (DA). The primary objective is to achieve high sensitivity while maintaining high performance across other metrics.

Figure 2 illustrates that the Conv Net-GAP model outperforms the other models in terms of overall performance. The proposed architecture seems to be effective for the task at hand, considering its superior performance compared to the baseline models. Additionally, the figure indicates that data augmentation (DA) plays a significant role in enhancing performance across most cases. This augmentation helps improve the model's ability to generalize and capture patterns effectively.

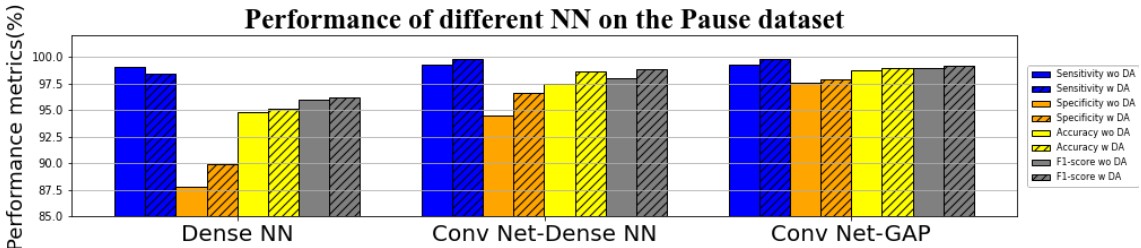

Figure 2: The accuracy and loss function during the training process in the training (blue line) and validation (green line) datasets using the proposed CNN

In the case of Dense NN or fully connected NN, DA may not have as significant an impact compared to CNNs. The reason why data augmentation may not have a substantial effect on fully connected

networks is related to the nature of the transformations applied during augmentation. Fully connected networks typically operate on flattened, vectorized representations of the input data, ignoring the spatial structure. Since fully connected networks lack the ability to exploit spatial locality, the potential benefits of data augmentation, which mainly aim to enhance the model's understanding of spatial relationships, may be limited. In contrast, CNNs are specifically designed to capture spatial hierarchies and patterns in input data through their convolutional and pooling layers. As a result, data augmentation can effectively enhance the performance of CNNs by introducing additional variations in the training data and improving the model's ability to generalize. In summary, data augmentation is more commonly applied and tends to have a more pronounced impact on convolutional neural networks due to their ability to capture spatial relationships and patterns in image data. Fully connected networks, on the other hand, are less influenced by data augmentation, as they do not directly leverage spatial information.

After identifying the optimal model based on training and validation datasets, we performed testing to assess its generalization performance. The training process for the proposed CNN is shown in Figure 3. The optimal threshold was chosen to obtain a relative sensitivity and specificity of 99.26% and 96.89%, respectively.

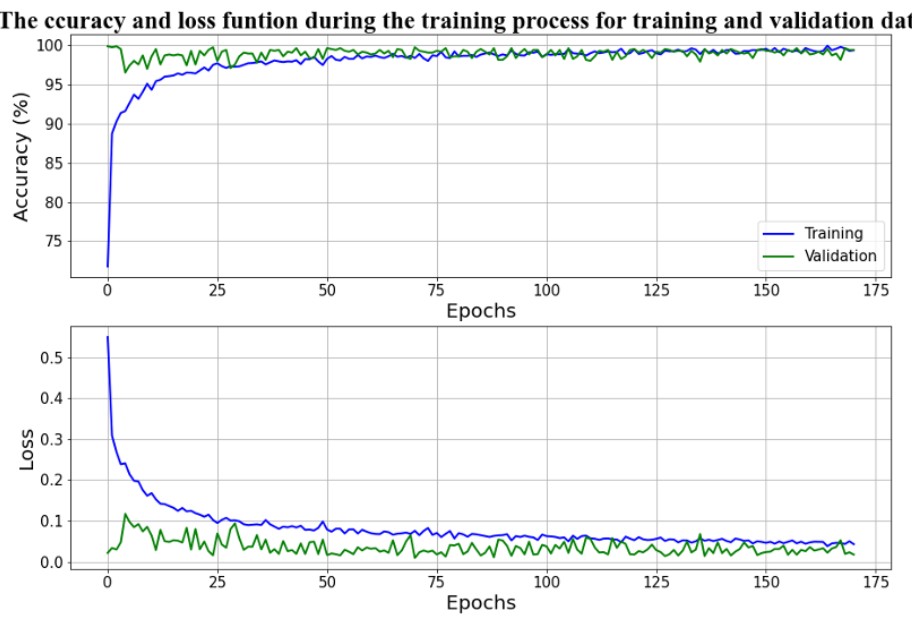

Figure 3: The accuracy and loss function during the training process in the training (blue line) and validation (green line) datasets using the proposed CNN

The independent patient test dataset from 750 patients included 1,986 episodes (744 true pause from 382 devices, and 1,242 false pause from 485 devices). Sensitivity, specificity, PPV, and NPV derived using the independent patient test dataset are shown in Table 1.

Figure 4 shows the sensitivity and 1-specificity curve as a function of the probability threshold in this independent patient test dataset. For the proposed CNN, an area under the curve (AUC) of 0.9942 was obtained, and a threshold of 0.65 was chosen with sensitivity and specificity of 99.06% and 95.17% respectively. Out of the 750 patients, 740 (98.67%) were successfully classified as false detection-free. Among the 485 patients initially identified with false positive EGMs, 468 (96.5%) were subsequently classified as false positive-free.

Prior to employing the proposed CNN model, the independent patient test dataset consisted of 1,986 episodes, with 744 true pauses detected from 382 devices and 1,242 false pauses from 485 devices PPV= 37.46% (Figure 5). After applying the proposed CNN model, 737 out of 744 episodes were correctly identified as pauses, resulting in a PPV of 92.47%. Consequently, there was a reduction of EGM burden by 59.87% (100 - ((60 + 737) / (744 + 1242))) (Figure 5).

Table 1: Performance metrics for the proposed CNN reported for the independent patient test dataset as raw proportion of episodes and the GEE estimates adjusting for multiple episodes per patient.

| METRICS | PROSPOSED CNN | PERCENTAGE(%) | 95% CI |
|---|---|---|---|
| Sensitivity | 737/744 | 99.06 | 98.07-99.62 |
| Specificity | 1182/1242 | 95.17 | 93.83-96.29 |
| PPV | 737/797 | 92.47 | 90.42-94.21 |
| NPV | 1182/1189 | 99.41 | 98.79-99.76 |
| Accuracy | 1919/1986 | 96.63 | 95.74-97.38 |

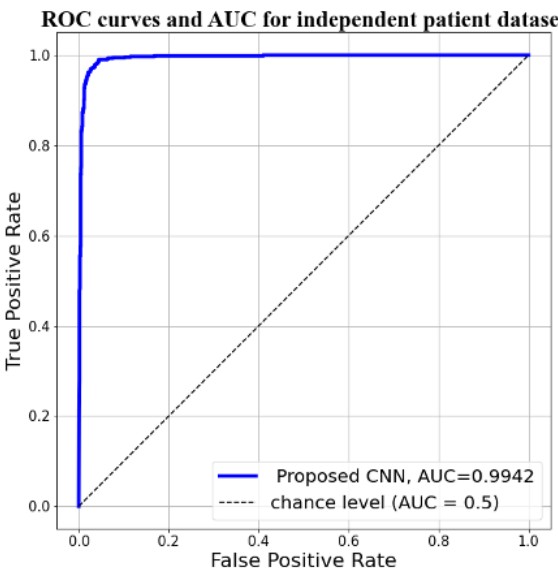

Figure 4: ROC curve and AUC as a function of the probability output from proposed CNN on independent patient test dataset.

## 4 DISCUSSION

Deep learning CNN is employed to classify Pause episodes detected by ICM. The objective was to reduce false detections while maintaining sensitivity for Pause detection. Instead of using classical machine learning techniques with manual feature extraction, an end-to-end approach was utilized, extracting features, and performing classification directly from the raw electrogram (EGM) signal. A 5-layer CNN was compared against baseline models, and the statistical results demonstrated a significant improvement.

In this study, a custom CNN was utilized with a total of 544,514 trainable parameters. This number is significantly lower compared to other pre-trained models such as RESNET18, which typically has more than 8 million parameters. By having a smaller number of parameters, the custom CNN in this study offers several advantages. First, it requires less computational resources for training and inference. Second, a smaller model size can help mitigate overfitting, a common challenge in deep learning. With fewer parameters to learn, the custom CNN may be less prone to overfitting the training data and therefore have improved generalization capabilities. This can be particularly beneficial when working with limited labeled data.

The performance of the deep learning network heavily relies on the consistency of the ICM ECG adjudication process. To ensure consistency, two reviewers were involved in the adjudication process. The adjudication process conducted by a third adjudicator was validated against adjudications performed by electrophysiologists using a small subset of the data, as described previously. The

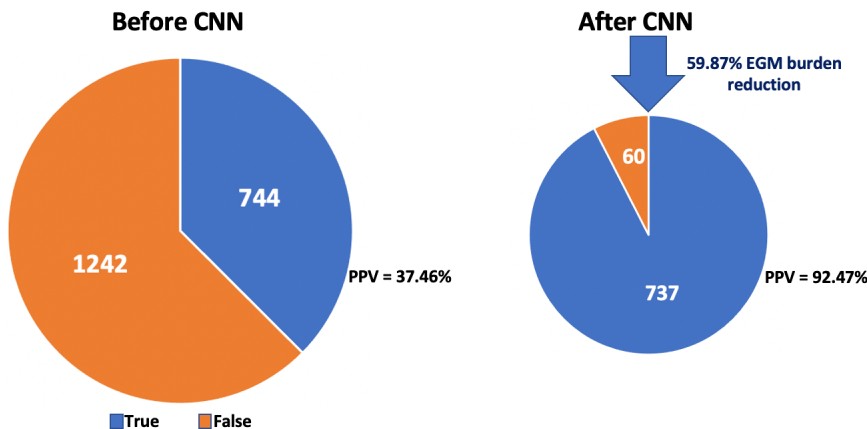

Figure 5: (Left pie) Before using proposed CNN, the independent patient test dataset included 1,986 episodes (744 true pause from 382 devices, and 1,242 false pause from 485 devices) has PPV=37.46%. (Right pie) After using proposed CNN, 737 episodes out of 744 correctly detected as pause has PPV=92.47%. We have 59.87% EGM burden reduction.

adjudicator's performance showed less than 1% error compared to the electrophysiologists' adjudications.

Furthermore, in the training and validation datasets, discordant analyses were conducted multiple times to identify reasons for mismatches between adjudication and CNN-predicted class labels. In cases where mismatches were attributed to manual errors, corrections were made to the erroneous adjudications. The independent test set was adjudicated twice by the same adjudicator, and any remaining mismatches were reviewed for a third time to establish the final adjudication. Moreover, increasing the size of the training dataset enhances the generalizability of any trained network. Data augmentation was employed with a magnification factor of 2, taking into account possible variations in the ICM device's positioning and movement, which could result in the presence of negative R-peaks in the dataset. To ensure transparency and reproducibility, the study reports results both with and without data augmentation for all baseline models, aiming to improve the quality and reproducibility of work in the field of data augmentation on deep learning-based ECG analysis.

Additionally, we are actively exploring the integration of transfer learning techniques Sun et al. (2022); Weimann & Conrad (2021) and personalized algorithms Yamaç et al. (2022) into these models, with the aim of further enhancing their efficacy and tailoring them to meet the unique needs of individual users.

The primary limitation of the study is that the deep learning network was trained using ECG data obtained from a specific device with a single-lead ECG vector and electrode separation of 4 cm, implanted at various locations and orientations. Therefore, the trained network's generalizability to other forms of ECG with different electrode configurations, such as 12-lead ECG systems, is limited. However, the same methodology can be applied to train a similar network using data collected from the chosen monitoring mode, thus enabling the potential for wider application and adaptability in the field.

The objective of this study was to develop specific CNN models tailored to reduce inappropriate pause episode detections in ICMs while preserving sensitivity, and evaluate the clinical performance of the models.

By leveraging CNN models and potentially deploying them within a cloud-based solution, the study sought to improve the accuracy and efficiency of pause detection in Abbott ICMs. This improvement would benefit both physicians and patients by facilitating more accurate diagnoses and effective management of cardiac arrhythmias.

## 5 CONCLUSION

A customized CNN model developed and tested in this study substantially reduced false pause episodes, with minimal impact on true pause episodes. The enhanced pause episode classification led to reduced overall episode review burden and much improved positive predictive value in the episodes to be reviewed by clinicians. Implementing the model on the manufacturer's patient care network may accelerate clinical workflow of ICM patient management.

### FUNDING SOURCES

The work was funded by Abbott

### DISCLOSURES

Authors from Abbott only need to disclose employment, physician authors need financial disclosures

### PATIENT CONSENT AND ETHICS STATEMENT

This study used fully deidentified data from Merlin.net™ patient care network, no ethics committee or institutional review board approval was necessary.

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
