# OpenReview forum: "Deep Learning-based Discrimination of Pause Episodes in Insertable Cardiac Monitors"
_ICLR.cc/2024/Conference — ICLR 2024 Conference Withdrawn Submission_

### Official Review · Reviewer_5Euk · 2023-10-27

**Soundness:** 1 poor
**Presentation:** 1 poor
**Contribution:** 1 poor
**Rating:** 1
**Confidence:** 5

**Summary:**

The authors employed a CNN model to classify ECG episodes from insertable cardiac monitors. The dataset included ECG episodes identified as pauses by an existing ICM system, which were subsequently manually labeled as true pauses or false pauses by annotators. The resulting CNN achieved an AUC of 0.9942 and an accuracy of 96.63%.

**Strengths:**

- The CNN model proposed by the authors achieved high performance, indicating that the task addressed in the paper can be effectively solved using deep learning methods.

**Weaknesses:**

- The methodology employed by the authors lacks novelty, as it follows the standard process of applying deep learning methodologies to address a specific task.

- The authors' review of related works appears to be insufficient. The reference section includes only 10 papers, which seems inadequate given the paper's length. Related works should encompass topics such as applying machine learning methods to ECG data from ICM devices, ECG pause detection algorithms, and other related areas.

- The implementation details lack sufficient explanation. To enable the reproduction of the experiments, specific aspects need further elaboration, including the size of the input data, the learning rate employed, the architecture of the models under comparison, the type of augmentation applied, and other relevant parameters.

- The organization of the paper requires improvement.
  - The labeling process, currently explained in the discussion section, appears more suitable for the dataset description rather than a discussion item.
  - There are duplicate sentences present on both page 3 and page 6 (Moreover, increasing the size of the training dataset …… deep learning-based ECG analysis).
  - There is a caption duplication issue where the caption for Figure 2 is the same as that of Figure 3.

- The arguments by the authors appear to lack supporting evidence.
  - “In summary, data augmentation is more commonly applied and tends to have more pronounced impact on convolutional neural networks due to their ability to capture spatial relationships and patterns in image data” on page 4: the authors should conduct comprehensive experiments that encompass various augmentation schemes and widely-used model architectures.
  - “With fewer parameters to learn, the custom CNN may be less prone to overfitting the training data and therefore have improved generalization capabilities” on page 5: The authors should experiment with networks of varying sizes and provide evidence to demonstrate that their selected network size is appropriate for this specific task.

**Questions:**

- The data used in the experiments is the collection of ECG episodes that were predicted as pause episodes by an existing ICM system. Why did the authors choose to develop a backup system for an existing algorithm, rather than directly addressing the pause detection problem using raw ECG signals? If a high-performing CNN can be directly applied to detect pause episodes from raw ECG data, it would eliminate the need for an extra step. Are there any practical reasons for this approach?

- To fully understand the task, it is essential to explain how an episode is defined. What is the duration of a single episode? Is it an ECG signal containing a brief pause within it? Providing a more detailed description of the task's input would be beneficial.

- How crucial are Sensitivity and Specificity in the context of this study? Which is more important: minimizing false positives or minimizing false negatives for achieving the goal of the proposed task?

---

### Official Review · Reviewer_zXt2 · 2023-10-29

**Soundness:** 3 good
**Presentation:** 2 fair
**Contribution:** 1 poor
**Rating:** 1
**Confidence:** 5

**Summary:**

The authors developed CNN model to discriminate pause episodes in insertable cardiac monitors.

**Strengths:**

The project seems prety much straight forward. However it is quite difficult to evaluate since both code and data are not available.

**Weaknesses:**

The scope and the direction of the study is not compatible with the ICML community. The authors should consider more specific venues.

**Questions:**

There are many typos and usage of abbrevuatuins without full spelling at the first appearance, e.g., EGM in the abstract.

---

### Official Review · Reviewer_cDD3 · 2023-11-09

**Soundness:** 2 fair
**Presentation:** 1 poor
**Contribution:** 1 poor
**Rating:** 1
**Confidence:** 5

**Summary:**

The paper presents a deep learning method for analysing ECG data from insertable cardiac monitors. The authors aim two detect pauses in ICMs data.

**Strengths:**

The authors propose utilising deep learning strategies for the analysis of ECG signals from wearable sensors and hence identify condition such as arrhythmias. The aim is to reduce the false negatives while maintaining a reasonable true positive performance.

**Weaknesses:**

The methods proposed are mostly already present in literature, CNNs are widely used to analyse ECG signals and most of the paper is dedicated to how the data was curated. It would have been more interesting to see how this method differs from other methods in literature in terms of how to deal with 1D ECG signals.

**Questions:**

It is suggested to look at the wide range of literature on using CNNs for ECG signal analysis and compare results.

Are the performance metrics indicating overfitting to the data?  a comprehensive answer will require an independent dataset or validation on some of the widely available benchmark datasets for cardiac arrhythmia such as those provided by physionet.